# Classroom Furniture Mismatch and Back Pain Among Adolescent School-Children in Abha City, Southwestern Saudi Arabia

**DOI:** 10.3390/ijerph16081395

**Published:** 2019-04-18

**Authors:** Abdullah Assiri, Ahmed A. Mahfouz, Nabil J. Awadalla, Ahmed Y. Abouelyazid, Medhat Shalaby, Ahmed Abogamal, Abdullah Alsabaani, Fatima Riaz

**Affiliations:** 1Department of Internal Medicine, College of Medicine, King Khalid University, Abha 61421, Saudi Arabia; aassiri@yahoo.com (A.A.); medhatshalaby57@gmail.com (M.S.); 2Department of Family and Community Medicine, College of Medicine, King Khalid University, Abha 61421, Saudi Arabia; njgirgis@yahoo.co.uk (N.J.A.); drzizous2000@yahoo.com (A.Y.A.); dr.alsabaani@hotmail.com (A.A.); fatima.riaz786@yahoo.com (F.R.); 3Department of Epidemiology, High Institute of Public Health, Alexandria University, Alexandria 21511, Egypt; 4Department of Community Medicine, College of Medicine Mansoura University, Mansoura 35516, Egypt; 5Department of Rheumatology, Faculty of Medicine, Al Azhar University, Cairo 11651, Egypt; ahmedfathy15@yahoo.com

**Keywords:** back pain, school children, classroom furniture, mismatch, ergonomics

## Abstract

*Objective*: To explore the potential classroom furniture mismatch with students’ anthropometric measurements and back pain related to sitting for extended periods. *Methods:* This cross-sectional study was carried out on all twelve male and female intermediate and secondary schools located in Abha city. Anthropometric and classroom furniture measurements were assessed, and the mismatch was determined using standardized methods. Students were also screened for back pain related to long sitting at school. *Results:* A total number of 879 students was selected. The study revealed seat height mismatch in both intermediate, and secondary school of 84.3%, and 75.6%, respectively. Seat depth mismatch was 74.0% in intermediate schools and reached 84.5% in secondary schools. The desk height was improper for 94.1%, and 82.3% of students in intermediate, and secondary schools, respectively. The levels of mismatch differ significantly by grade level and gender. A prevalence of 10.8% of back pain related to long sitting at school was found. In multivariable logistic regression, males, intermediate school children, and the presence of buttock-popliteal length/seat depth mismatch were significantly associated with pain. On the other hand, practicing exercise was a significant protective factor. *Conclusions:* There is a prevalent mismatch between students’ dimensions and existing schools’ furniture. School furniture providers should take in consideration the average Saudi students’ dimensions, while designing school furniture especially for males, and at intermediate schools, or provide schools with adjustable seats and desks.

## 1. Introduction

School children attend schools for a considerable amount of time, 6 h a day on average. They spend the largest part of their time at school in performing different related activities, which lead them to be seated on their seats constantly for a long time [1].

Literature review has shown that school children often use furniture that does not match their anthropometric requirements [2,3]. This has been recognized by the fact that school furniture dimensions are usually based on findings not correlated with the students’ requirements [4]. School children are growing and changing in dimensions. This results in high variability in body dimensions in the same school grade. Ergonomic mismatch develops when several subjects of different anthropometric dimensions use a fixed one-size-fits-all furniture [5].

Students are especially predisposed of negative health impacts from inadequately designed and mismatched furniture, due to prolonged periods of sitting. Furthermore, improper school furniture may also be accountable for poor sitting posture. Improper students’ posture, while using school furniture, is considered one of the factors that may increase the possibility of developing back pain [6]. Back pain among school children is a highly prevalent problem, which could influence their learning capabilities [2] and substantially impact their adolescent and adulthood quality of life [7]. The reasons for the development of back pain remains debatable, and mainly relates to musculoskeletal strain associated with bad ergonomics during daily activities [8,9]. The use of ergonomically matched school furniture may reduce children’s fatigue and back pain in the school environment [10].

Ergonomic data regarding school furniture in Abha city, south western Saudi Arabia, are scarce. Also, there is a lack in literature regarding back pain related to long periods of sitting at school and its relationship with school furniture mismatch. The objectives of the present study were to investigate whether the current school furniture, which were provided by school authorities, match with anthropometric data of adolescent children and to explore the relation with back pain related to long sitting.

## 2. Methods

### 2.1. Study Design

The study design was a cross-sectional study.

### 2.2. Study Settings

The study was conducted on all twelve male and female intermediate and secondary schools located in Abha city, Asser Region, Southwestern Saudi Arabia. The age of the intermediate and secondary grades students in Saudi Arabia range from 13–15 years, and 16–18 years, respectively.

One class was randomly chosen from each of the three levels in each school. Consent letters were sent to the guardians of all children in each of the selected classes. During the field visit, all present children with approval from their parents were examined. All procedures performed in studies involving human participants were in accordance with the ethical standards of the institutional and/or national research committee and with the 1964 Helsinki Declaration and its later amendments or comparable ethical standards. The research proposal was revised and accepted by the ethical committee of King Khalid University (REC # 2014-03-09). Approval was also obtained from school authorities.

### 2.3. Anthropometric Measurements

Student measurements (except height) were assessed while setting erect on a flat horizontal surface, with knees bent 90°, and without shoes. Height was measured while standing erect without shoes. The following body dimensions were assessed [5,11,12]:*Elbow height*. The distance from the student’s seated surface to his olecranon (bottom of the elbow tip) was measured while flexing the elbow at 90°.*Knee height.* The vertical distance from the foot resting surface to the top of the knee cap just in back and above the patella was measured with the knee flexed at 90°.*Popliteal height.* The vertical distance from the posterior surface of the knee or popliteal space to the foot resting surface was measured with the knee flexed at 90°.*Buttock-popliteal length*. The horizontal distance from the posterior surface of the knee or popliteal space to the posterior surface of the buttock was measured with the knee flexed at 90°.*Height.* While the student stands upright, looking straight forward, the vertical distance from the floor to the top of the head was measured.Children’s weight was measured in kg, using standard methods, while the child wearing minimal clothes.

### 2.4. Classroom Furniture Dimensions’ Measurements

For each school, their classrooms’ desk/seat furniture were inspected, and their dimensions were measured. The following classroom furniture measurements were assessed to be used for calculating student’s furniture mismatch [5,11,12]:*Seat height* is the vertical distance from the highest point on the front of the seat to the floor.*Seat depth.* The chair seat depth is the horizontal distance of the sitting surface from the back of the seat, at a point where it is assumed that the buttocks begins, to the front of the seat.*Desk/table height*. The desk/table height is the vertical distance from the floor to the top of the front edge of the desk or table.

Assessment of ergonomic mismatch [5,11,12]:*Popliteal Height and Seat Height Mismatch.* Popliteal and seat height as any seat height that was either >95% or <88% of the popliteal height.*Buttock-Popliteal Length and Seat Depth Mismatch.* A mismatch of buttock-popliteal length to seat depth was defined as a seat that is either <80% or > 95% of the buttock-popliteal length.*Knee Height and Desk/Table Height Mismatch.* Desk–knee clearance should exceed 2 cm. Thus, a mismatch is defined to occur when a desk/table is <2 cm higher than the knee height.*Elbow Rest Height and Desk/Table Height Mismatch.* Mismatching was considered when the following matching equation was not fulfilled: Elbow height ≤ Desk height ≤ Elbow height + 5 cm

Intensive training was performed for the data collectors to standardize methods of measurements and to minimize inter and intra observer variations.

### 2.5. Back Pain

Students were screened for back pain related to long sitting at school. The daily schedule for students is usually 6 classes per day. Each class lasts for 50 min. Students are usually sitting all the time of the class. Only five minutes break were allowed between classes. During the day there is a single break for 15 min.

Back pain was assessed using the following question: “During the past three months have you felt any pain or ache in the back area while sitting for a long time at school, which has lasted for one day or longer?” The response alternatives were “yes”, “no” or “I don’t know”. The last option was regarded as “no”. Screening for back pain related to long sitting at school was selected because it would be a proxy for the health effect of school furniture mismatch, if any.

### 2.6. Data Analysis

Data entry and analysis was done using SPSS program version 22 (IBM Corporation, Armonk, NY, USA). The student’s *t*-test was used to compare anthropometric measurements between boys and girls in intermediate and secondary schools. Also, chi-square test was used to evaluate the association between gender and levels of mismatch. P value of less than 5% was used as level of significance. Proportions with 95% confidence intervals (95% CI) were used to present the prevalence of back pain. Multivariable binary logistic regression analysis was used to identify the independent factors associated with back pain related to long sitting at school. Adjusted odds ratios (aOR) and 95% confidence intervals were calculated. Variables included in the model were, gender, school grade, physical exercise (any regular engagement in physical activities like running, cycling, swimming, brisk walking, playing football, etc.), over weight and obesity, and classroom furniture mismatches.

## 3. Results

Twelve boys’ and girls’ schools were included in the current study; six intermediate and six secondary schools in Abha City. The study included 879 students (439 intermediate grade students and 440 secondary grade students). The mean and standard deviation values of the student’s anthropometric measurements by grade and gender were presented in Table 1. The table shows that the average body mass index (BMI) values were significantly higher (*p* = 0.01) among boys (23.87 ± 6.48 kg/m^2^) compared to girls (22.24 ± 4.27 kg/m^2^) in the secondary grade. On the other hand, the BMI values were not significantly different among boys and girls in the intermediate grade.

Regarding other anthropometric measurement, the table shows that popliteal height in intermediate grade was significantly (*p* = 0.001) higher among boys (43.10 ± 2.50 cm) compared to girls (41.40 ± 2.46 cm). A similar trend was observed in intermediate and secondary grades where average boys’ popliteal height (in cm), buttock popliteal length (in cm) and knee-height (in cm) were significantly higher compared to girls. Similarly, in secondary grade, the average values for elbow height (in cm) among boys was significantly higher compared to girls. Meanwhile, no significant difference was observed in intermediate grade.

Table 2 shows ergonomic assessment of student’s furniture match/mismatch percentage by gender and grade level. Overall, most of the students in intermediate and secondary schools showed mismatch in the three studied measurements (Popliteal Height/seat height, Buttock-popliteal length/seat depth and elbow rest height/desk height). The overall seat height mismatch in both intermediate and secondary school reached 84.3%, and 75.6%, respectively. The seat depth mismatch was 74.0% in intermediate schools and reached 84.5% in secondary schools. The desk height was improper for 94.1% and 82.3% of students in intermediate and secondary schools respectively. The majority of mismatch in seat and desk heights showed a high mismatch (too high seat and desk). On the other hand, most of mismatch in seat depth was low mismatch indicating a shallow seat (data not tabulated). It was found that none of the students experienced knee clearance problem as the knee-desk distance exceeded 2 cm height for all the students in both grades.

Regarding gender differences, Table 2 shows that there were significant differences in the percent of mismatch, where males had a higher buttock popliteal length/seat depth in intermediate and secondary, and popliteal height in secondary schools only. On the other hand, the table shows a significant difference where females had higher values in knee height/desk height in intermediate and secondary grades.

The study showed that 95 students complained of back pain related to long sitting at school giving a prevalence of 10.8% (95% CI: 9.2 %–12.3%).

Table 3 shows factors associated with back pain related to long sitting at school. Males were significantly having back pain compared to females (aOR = 2.101, 95% CI: 1.218–3.625). Similarly, intermediate school children were significantly having more back pain (aOR = 2.242, 95% CI: 1.364–3.685) compared to secondary school students. The only significant mismatch measurement associated with back pain was buttock-popliteal length / seat depth (aOR = 3.386, 95% CI: 1.403–8.173). On the other hand, the study showed that undertaking exercise was a significant protective factor for back pain (aOR = 0.640, 95% CI: 0.405–0.901).

## 4. Discussion

Overall, the majority of students in intermediate and secondary schools showed mismatch in the three studied measurements (Popliteal Height/seat height, Buttock-popliteal length/seat depth and knee height/desk height). These results were in accordance with Iranian study [12], which reported considerable mismatch between body dimensions of the school students and the classroom furniture. Also, another Indonesian study reported a classroom furniture standards/student’s dimensions mismatch values ranging from 63.4% to 96.7% [13]. Similar results were observed in Chilean elementary schools [3] and Michigan schools [5]. The high mismatch percentage in the present study between furniture and students’ anthropometry could be explained by the fact that the purchasing of the furniture was made without taking in consideration the ergonomic principles and the variability of students’ dimensions within the same grade level and across the different grades. These conditions may increase the risk of back pain and spinal problems [12].

Interestingly, the current study observed that the majority of mismatch in seat and desk heights were high mismatch. On the other hand, most of mismatch in seat depth was low mismatch. This indicates that most of the students are using too high and too narrow chairs. The same results were reported in Iranian study [14].

On examining the relation between students’ grade and schoolroom furniture mismatch, the present study detected that mismatch significantly decreased with increased grade level for seat height and disk height while significantly increased for seat depth. This could be explained by the fact that high mismatch (high desk) decrease with students’ growth. On the other hand, the low mismatch in seat depth (narrow chair) increase with students’ growth. The same findings were observed in the Iranian study [12].

Regarding gender differences, the present study observed that boys had a higher seat depth mismatch (narrow seat) in intermediate and secondary, and seat height mismatch (high seat) in secondary schools only compared to girls. On the other hand, the girls had higher mismatch values of desk height in intermediate and secondary grades compared to boys (high desk). The gender difference in mismatch was observed in other previous studies [5,10,12,13,15]. The effect of gender on frequency of mismatch could be attributed to the developmental differences at the time of puberty between boys and girls at each grades [5].

The current study revealed that 10.8% of school children were suffering back pain related to long sitting at school. The back pain related to prolonged sitting may be due to sitting with improper posture or using school furniture that mismatching the anthropometric measurements of school children [9,16]. Back muscles strain resulted from exertions used to maintain balance and comfort while sitting on improperly designed school chairs is a causal factor for back pain at schools [6,10]. Furthermore, the instability resulted from mismatched school furniture forced the students to take improper postures while sitting, which consequently exaggerating the back-pain problem [5,9,15]. The result of the present study endorses these findings as the buttock-popliteal length/seat depth mismatch was an independent factor associated with long sitting back pain at school. the use of wider seat depth than buttock-popliteal length of the student may interfere with the use of the backrest of the seat and increase the pressure on students’ thighs [14,17]. Also, the use of narrow chairs may interfere with thigh support during sitting causing instability and discomfort [18]. Another independent factor for back pain in the present study was being in intermediate school grade. This could be explained by the fast-growing body dimensions during this age period and the subsequent highly prevalent desk/chair mismatches.

The present study revealed that the risk of back pain related to sitting was higher among male students compared with females. Posture during sitting differs significantly by gender. Females tend to take more comfortable posture with back rest compared with males. This posture places males at a greater risk of developing back pain and disc herniation [19,20,21].

The current study endorses the protective effect of practicing exercise in minimizing back pain related to long sitting at school. This is in agreement with previous studies [22,23,24]. The possible explanation of this beneficial effect is that regular exercises increase intercostal and back muscles strength and consequently decrease frequency and intensity of back pain [25].

### Limitations

The outcomes of the current study should be considered in light of some study limitations. The first limitation was that the samples were taken from one city in southwestern region of Saudi Arabia. Therefore, it is possible that the selected schools in the Abha city are not representative of other areas of the Saudi Arabia. Around 97.8% of the study samples were Saudis. Taking other nationalities into consideration is likely to provide small numbers. Therefore, the effect of students’ nationality was not considered during analysis. Anthropometric dimensions of the school children could be affected by student’ nationality. Another study limitation was a lack of assessment of the students’ posture during sitting.

## 5. Conclusions

There is a considerable mismatch between the body dimensions of the students and the existing classroom furniture. Most of the mismatch was a high mismatch for seat and desk heights and low mismatch for seat depth. This indicated that most of the students in the intermediate and secondary schools in Abha, southwestern Saudi Arabia, are sitting on too high, on too narrow chairs, and improper desk height. The levels of mismatch differ by students’ grade level and gender. These possibly resulted from different growth rates and furniture measurements. Back pain related to long sitting at school is evident. The risk is higher among males, at intermediate schools and in the presence of improper chairs dimensions. Practicing physical exercise has a beneficial effect in minimizing back pain. The findings of the study suggest that, the manufacturers should take into consideration the Saudi students’ dimensions, while designing school furniture. It is recommended that a nation-wide study should be established to assess the diversity of mismatch in terms of percentiles to help designers. Another suggestion is to provide schools with adjustable seats and desks. Some studies recommended adjustable school furniture to improve students’ posture, learning capabilities, and reduce the risk of musculoskeletal complaints [26,27,28]. It is also highly recommended to encourage practicing physical exercises among school children.

## Figures and Tables

**Table 1 ijerph-16-01395-t001:** Anthropometric measurements of students participated in the study (*n* = 879).

Anthropometric Measurements	Intermediate				Secondary			
	All (*n* = 439) Mean (SD)	Boys (*n* = 145) Mean (SD)	Girls (*n* = 294) Mean (SD)	*p*-Value	All (*n* = 440) Mean (SD)	Boys (*n* = 329) Mean (SD)	Girls (*n* = 111) Mean (SD)	*p*-Value
Weight (Kg)	54.47 (16.13)	58.97 (19.10)	52.24 (13.95)	0.001	64.02 (18.68)	67.25 (19.37)	54.75 (12.60)	0.001
Height (cm)	157.31 (8.95)	163.13 (10.51)	154.44 (6.36)	0.001	164.77 (9.30)	167.62 (8.02)	156.46 (7.67)	0.001
Body Mass Index (kg/m^2^)	21.82 (5.42)	21.86 (5.85)	21.80 (5.21)	0.91	23.45 (6.03)	23.87 (6.48)	22.24 (4.27)	0.01
Popliteal Height (cm)	41.97 (2.60)	43.10 (2.50)	41.40 (2.46)	0.001	43.67 (7.08)	45.17 (6.12)	41.17 (2.55)	0.001
Buttock-Popliteal Length (cm)	52.37 (5.21)	56.86 (4.75)	50.15 (3.81)	0.001	54.72 (6.39)	56.64 (5.07)	50.08 (3.36)	0.001
Knee-Height (cm)	49.00 (4.43)	51.48 (3.66)	48.01 (2.60)	0.001	51.70 (5.69)	53.23 (3.67)	48.31 (2.90)	0.001
Elbow Height (cm)	61.32 (4.52)	61.58 (4.86)	61.33 (3.65)	0.54	64.80 (5.05)	66.09 (3.53)	61.48 (4.17)	0.001

*p*-Values: comparing boys and girls.

**Table 2 ijerph-16-01395-t002:** Students- furniture match/mismatch percentages by gender and grade level (*n* = 879).

Grade/Gender	Ergonomic Assessment					
	Popliteal Height/Seat Height		Buttock-Popliteal Length/Seat Depth		Elbow Rest Height/Desk Height	
	Match No. (%)	Mismatch No. (%)	Match No. (%)	Mismatch No. (%)	Match No. (%)	Mismatch No. (%)
**Intermediate**						
Boys (*n* = 145)	22 (15.2)	123 (84.8)	9 (6.2)	136 (93.8)	20 (13.8)	125 (86.2)
Girls (*n* = 294)	47 (16.0)	247 (84.0)	105 (35.7)	189 (64.3)	6 (2.0)	288 (98.0)
Total (*n* = 439)	69 (15.7)	370 (84.3)	114 (26.0)	325 (74.0)	26 (5.9)	412 (94.1)
*p*-value	0.890		0.001		0.001	
**Secondary**						
Boys (*n* = 329)	63 (19.2)	266 (80.8)	28(8.5)	301 (91.5)	68 (20.7)	261 (79.3)
Girls (*n* = 111)	44 (39.6)	67 (60.4)	40 (36.0)	71 (64.0)	3 (2.7)	108 (97.3)
Total (*n* = 440)	107 (24.4)	333 (75.6)	68 (15.5)	372 (84.5)	71 (16.1)	369 (83.9)
*p*-value	0.001		0.001		0.001	

**Table 3 ijerph-16-01395-t003:** Personal and school furniture mismatch factors associated with back pain related to long sitting at school among the study sample of adolescents (*n* = 876).

Factors		aOR (95% CI)
Gender	Female	Ref
	Male	2.101 (1.218–3.625)
Undertaking exercise	No	Ref
	yes	0.64 (0.405–0.901)
School grade	Secondary	Ref
	Intermediate	2.242 (1.364–3.685)
BMI (kg/m^2^)	18.5–24.9	Ref
	25 and more	0.766 (0.457–1.283)
Popliteal Height/Seat Height mismatch	No	Ref
	yes	0.782 (0.453–1.350)
Buttock-popliteal length/Seat Depth mismatch	No	Ref
	yes	3.386 (1.403–8.173)
Elbow rest height/Desk Height mismatch	No	Ref
	yes	0.753 (0.389–1.456)

aOR: adjusted odds ratios.

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
