# Peer review of "Classroom Furniture Mismatch and Back Pain Among Adolescent School-Children in Abha City, Southwestern Saudi Arabia"

_ijerph, 2019, doi:10.3390/ijerph16081395_

Round 1

Reviewer 1 Report

It is a good paper, but I have a few improvements to mention:

1

at row 111 it is not clear if +5 is in mm or cm

2

at row 112 long sitting is not defined: please explain how many hours the children are sitting per day and how many breaks in minutes they have

3

about units specification: please use mm as smallest numbers, no digits necessary because if you measure a child twice repeately, you will measure at least 5 mm off at most dimensions

so height can be 1573 mm but 157.31 to too precise for this type of anthropometry

also BMI 21.8 (5.4) kg is enough precision and 21.82(5.42) is not necessary

4

Next time for you as researcher please report the outcome of the repeatabily session so the reader can see how precise you can measure; this means measure a child twice by the sam

e researcher with the same instrument within a short time

5

Because intermediate and secundary grade might differ per culture, its better to also  describe the age distribtuion of the subjects in table 16

6

I also would recommend to include a few pictures of the measures schoolchildren using the furniture to give the reader a good impression of the situation. For reasons of ethics you might want to blur the faces

7

for designers who want to use your data in this publication I would recommend to include also P3 and P97 of each variable you measured. because its is better to design for diversity P3-P97 or better P1-P99 than for the mean, because if you design a doorheight for the mean, 50 % will bump their heads.

8

For reasons of easy understanding I would recommend to include a few pictures that show the worst missmatch for each dimension

Author Response

The authors would like to acknowledge the constructive comments of the reviewer. Changes are made in red. Please find point by point response to the raised comments:

1.at row 111 it is not clear if +5 is in mm or cm

·        At row 112 it should read +5 cm.

2. at row 112 long sitting is not defined: please explain how many hours the children are sitting per day and how many breaks in minutes they have·        Definition of long sitting was mentioned in line 117 as follows “The daily schedule for students is usually 6 classes per day. Each class last for 50 minutes. Students are usually sitting all the time of the class. Only five minutes break were allowed between classes. During the day there is a single break for 15 minutes”.

3. about units specification: please use mm as smallest numbers, no digits necessary because if you measure a child twice repeately, you will measure at least 5 mm off at most dimensions

so height can be 1573 mm but 157.31 to too precise for this type of anthropometry

also BMI 21.8 (5.4) kg is enough precision and 21.82(5.42) is not necessary

·        About units’ specification, the figures presented in the results and tables were mostly average and standard deviation values. Individual measurements were taken to the nearest figure, yet, the calculated mean and standard deviations included decimals and should be reported as such.

4. Next time for you as researcher please report the outcome of the repeatabily session so the reader can see how precise you can measure; this means measure a child twice by the sam

e researcher with the same instrument within a short time

·        Regarding the issue of outcome of the repeatably sessions, “Intensive training was performed for the data collectors to standardize methods of measurements and to minimize inter and intra observer variations.” Line 113

5. Because intermediate and secundary grade might differ per culture, its better to also  describe the age distribtuion of the subjects in table 16

·        The issue of the age of the intermediate and secondary grade in Saudi Arabia was mentioned in line 69.

6. I also would recommend to include a few pictures of the measures schoolchildren using the furniture to give the reader a good impression of the situation. For reasons of ethics you might want to blur the faces

·        Regarding the issue of including pictures, for ethical and cultural reasons, especially with no previous consent taken from students and their parents, authors feel that this issue will raise potential problems.

7. for designers who want to use your data in this publication I would recommend to include also P3 and P97 of each variable you measured. because its is better to design for diversity P3-P97 or better P1-P99 than for the mean, because if you design a doorheight for the mean, 50 % will bump their heads.

·        The objectives of the present work were to study relations between back pain and current ergonomic mismatch. Further recommendation to be added is to start a nation-wide study to assess the diversity of mismatch in terms of percentiles to help designers in the future. Line 253

Reviewer 2 Report

Thank you for the opportunity to review this paper. My comments are as follows:

Abstract: Appropriate.

Introduction: Line 40, "School children used to live at their schools for a considerable fraction of their time.." should be "School children attend school for a considerable amount of time..."

Some issues with English throughout.

Parental consent appears to be obtained for students. Was approval obtained from a local school board or ethics committee? This should be stated and if not obtained the reason why ethics approval was not required should be stated (particularly as the research involves students).

Methods: Minor issues with English impact on clarity. What exercise were students asked about? This does not appear to be described in the methods.

Results: Minor issues with English impact on clarity e.g. page 6 line 154-55 "Majority of mismatch in seat and desk heights were high mismatch indicating too high seat and desk."

Line 164-165 repeats 156-157.

Line 173 - "practicing exercise" should be "undertaking exercise" 

Did the authors consider using age in the analysis as a proxy for development stage?

It is unclear as to the type of exercise. This would be of use for the reader particularly as it has a protective effect.

Discussion: Line 222-23, "Posture during sitting differs significantly by gender. Females tend to take more comfortable posture with back rest compared with males." Is there evidence of this in the current study?

Limitations section: Are a range of nationalities likely to be present within the school sample? If a range of nationalities are present within schools why was this not assessed. Or are the authors suggesting that Saudi students may be different from those in other countries?

Has there been a change in growth patterns (i.e. are students taller now compared to previous times?).

Some minor issues with English that impact on clarity.

Tables are appropriate.

Author Response

The authors would like to acknowledge the constructive comments of the reviewer. Changes are made in red. Please find point by point response to the raised comments:

Introduction: Line 40, "School children used to live at their schools for a considerable fraction of their time.." should be "School children attend school for a considerable amount of time..."·        Line 40 should read “School children attend school for a considerable amount of time”

Parental consent appears to be obtained for students. Was approval obtained from a local school board or ethics committee? This should be stated and if not obtained the reason why ethics approval was not required should be stated (particularly as the research involves students).

·        Line 262, approval was also obtained from school authorities.

Methods: Minor issues with English impact on clarity. What exercise were students asked about? This does not appear to be described in the methods.

·        Physical exercise was defined as “any regular engagement in physical activities like running, cycling, swimming, brisk walking,  playing football” Line 135

Results: Minor issues with English impact on clarity e.g. page 6 line 154-55 "Majority of mismatch in seat and desk heights were high mismatch indicating too high seat and desk."

·         Line 163 should read “Majority of mismatch in seat and desk heights were high mismatch (too high seat and desk).”

Line 164-165 repeats 156-157.

·        Regarding repeating statement “It was found that none of the students experienced knee clearance problem as the knee-desk distance exceeded 2 cm height for all the students in both grades. “. The second statement was omitted. The first statement was kept as it is (line 165)

Line 173 - "practicing exercise" should be "undertaking exercise"
·        The word practicing was replaced by the word “undertaking “line 180 and table 3”

Did the authors consider using age in the analysis as a proxy for development stage?

·        The age was considered in the analysis in an indirect way.  “The age of the intermediate and secondary grades students in Saudi Arabia range from 13-15 years and 16-18 years, respectively. “line 69”

It is unclear as to the type of exercise. This would be of use for the reader particularly as it has a protective effect.

·        The type of physical exercise was mentioned in details in line 135.

Discussion: Line 222-23, "Posture during sitting differs significantly by gender. Females tend to take more comfortable posture with back rest compared with males." Is there evidence of this in the current study?

·        Posture of siting of both genders was not assessed in the present study.

Limitations section: Are a range of nationalities likely to be present within the school sample? If a range of nationalities are present within schools why was this not assessed. Or are the authors suggesting that Saudi students may be different from those in other countries?

·        Regarding nationalities issue, the vast majority of the study sample students were Saudis (860, 97.8%).

Has there been a change in growth patterns (i.e. are students taller now compared to previous times?).

·        Growth pattern changes over time was not assessed in the present cross-sectional study. Yet, furniture was the same for different age groups (3 years period in each grade).

Round 2

Reviewer 2 Report

I thank the authors for addressing the previous comments.

Abstract: Appropriate.

Introduction: Line 40 should be "Schoolchildren attend school.."

Methods: Appropriate.

Results: Clearer.

Discussion: Limitations section: Are a range of nationalities likely to be present within the school sample? If a range of nationalities are present within schools why was this not assessed. Or are the authors suggesting that Saudi students may be different from those in other countries?

This comment is not really addressed by the authors. The "limitation" could be answered by the response above - the authors could state that 97.8% were Saudi, thus an assessment of other nationalities is likely to provide small numbers (or a similar statement).

Line 230-231 "Posture during sitting differs significantly by gender. Females tend to take more comfortable posture..." etc. If posture was not assessed, why isn't this a limitation?

Some minor issues with English remain in the manuscript.

Author Response

The authors would like to thank the reviewer for the comments. Changes are in red.

·        Line 39 was rephrased to read “School children attend schools”.

·        Line 235 should read “Around 97.8% of the study sample were Saudis. Taking other nationalities into consideration is likely to provide small numbers. Therefore, the effect of students’ nationality was not considered during analysis”.

·        Line 238 should read “Another study limitation was lack of assessment of the students’ posture during sitting”